# AggLCF: Aggregation Enhanced Localized Conformal Factuality for Large Language Models

## Abstract

With the growing generative capabilities of large language models (LLMs) in question answering, their practical deployment is hindered by unreliable outputs. Conformal methods have been introduced to control sub-claim factuality with theoretical guarantees. In this paper, we propose AggLCF, a **L**ocalized **C**onformal **F**actuality framework enhanced by multi-model **Agg**regation with rigorous factuality rate control. By semantically clustering diverse responses from multiple LLMs and extracting structured features, AggLCF learns a localized threshold to construct a filtered set of sub-claims that ensures local factuality while preserving more outputs through aggregation. AggLCF outperforms the previous state-of-the-art in conditional conformal methods on the MedLFQA benchmark with the highest number of retained valid sub-claims.

## 1 Introduction

The rapid advancement of autoregressive generative pre-training techniques (Vaswani et al., 2017) has propelled large language models (LLMs) such as ChatGPT (Achiam et al., 2023), Claude (Anthropic, 2024), LLaMA (Touvron et al., 2023), and Deepseek (Liu et al., 2024; Guo et al., 2025) from academic research into broad industrial applications. These LLMs with billions or even hundreds of billions of parameters, leveraging transfer learning capabilities from massive general-purpose corpora, can not only accomplish fundamental language tasks such as text completion, summarization, QA dialogue, and code generation, but also demonstrate fluency and diversity in more advanced scenarios like creative writing, knowledge retrieval, and knowledge integration. However, LLMs are prone to the phenomenon of *hallucination* (Ji et al., 2023; Nadeau et al., 2024; Varshney et al., 2023; Manakul et al., 2023b), wherein they generate semantically plausible but factually ungrounded or internally inconsistent content. This poses particularly critical challenges in high-risk domains such as finance (Wu et al., 2023), healthcare (Li et al., 2023), and legal applications (Shi et al., 2024), where unreliable outputs may lead to erroneous decisions, compliance failures, or legal risks, underscoring the urgent need for "Trustworthy AI".

To mitigate hallucination, both academia and industry have proposed various methods for uncertainty quantification and factuality guarantee of LLM outputs, including confidence-based posterior adjustment (Kuleshov et al., 2018), temperature scaling Guo et al. (2017), deep ensembles (Lakshminarayanan et al., 2017), Bayesian entropy estimation (Gal & Ghahramani, 2016), retrieval-augmented generation (Lewis et al., 2020, RAG), and self-consistency checks via natural language inference (Manakul et al., 2023a, SelfCheckGPT). However, for black-box models like LLMs, these methods have fundamental limitations of inability to access model parameters, difficulty in conducting structural-level interventions, and lack of theoretical factuality guarantees.

Conformal prediction (Vovk et al., 2005; Lei et al., 2018) has gained increasing attention in the last decade, particularly with the rise of black-box models. Its distribution-free and finite-sample coverage guarantee provides a robust framework for uncertainty quantification. Recent works have adapted conformal methods to LLMs such as *conformal factuality* (Mohri & Hashimoto, 2024), which filters sub-claims from LLM responses via a unified confidence threshold. CF ensures the factuality rate is bounded by a predefined level $\alpha$ marginally. However, the global threshold, chosen under a worst-case scenario, overlooks the needs of localized or domain-specific error control

Table 1: Comparison of outputs filtered with Conformal Factuality (CF), Conditional Boosting (CB), and Aggregation enhanced Localized Conformal Factuality (AggLCF). The source model of filtered sub-claims is in different colors.

| Question: | Does Vitamin D Help polymorphic light eruptions? |
|---|---|
| **CF** | Vitamin D has immunomodulatory effects and plays a role in skin health; some studies suggest that it may help reduce inflammation and improve skin barrier function, potentially benefiting individuals with PMLE. |
| **CB** | Polymorphic light eruption (PMLE) is a common skin condition triggered by exposure to sunlight; Vitamin D has immunomodulatory effects and plays a role in skin health, and some studies suggest it may help reduce inflammation and improve skin barrier function, potentially benefiting individuals with PMLE. |
| **AggLCF** | Polymorphic light eruption (PMLE) is a skin condition characterized by an itchy, blistering, or papular rash that occurs in response to sunlight exposure; Vitamin D may help mitigate symptoms of polymorphic light eruption (PMLE) due to its anti-inflammatory and immunomodulatory effects; Some studies suggest Vitamin D may help reduce inflammation and improve skin barrier function, potentially benefiting individuals with PMLE; Vitamin D levels should be checked in PMLE patients to determine if supplementation is necessary, and combination therapy with photoprotective measures or topical corticosteroids may enhance effectiveness. |

■ llama2-13b   ■ gpt-3.5-turbo   ■ qwq-32b

across different inputs (e.g., varying question types, contexts, or difficulty levels) and is often overly conservative. Building upon the conditional conformal prediction framework (Gibbs et al., 2025), Cherian et al. (2024) (denoted as CC) provides greater claim retention by learning adaptive confidence thresholds that optimize the number of retained sub-claims for different types of queries or sub-tasks. Although CC retains more valid information while maintaining approximate conditional factuality guarantee, its performance is sensitive to function class selection when estimating the conditional quantile, and its reliance on recomputing conformal predictions for conditional boosting to improve sub-claim retention rate introduces significant computational overhead.

Another way to improve the sub-claim retention is to provide diverse responses for sub-claim decomposition at the beginning. The diversity of responses from one LLM is often limited by its generative capability, and both Mohri & Hashimoto (2024) and Cherian et al. (2024) decompose sub-claims generated from a single LLM. This motivates us to enrich the diversity of responses to the same question by aggregating different responses generated in parallel from multiple LLMs. Unlike numerous queries from the same LLM, the diversity of different LLMs not only enlarges the coverage of potential claims but also provides robust performance on questions from various domains. On the other hand, multi-LLMs offer helpful information on sub-claims while constructing reliable features for them. Intuitively, a sub-claim expressed similarly by multiple LLMs should be assigned a higher confidence level.

In this paper, we propose a lightweight framework that aggregates responses from multiple LLMs via semantic clustering and provides localized conformal factuality for aggregation-enhanced sub-claims. Benefiting from the semantically aggregation of enriched sub-claims generated in parallel by multi-models, our method increases the number of retained sub-claims while delivering robust localized conformal factuality with rigorous marginal guarantees. Different from Cherian et al. (2024), our method adaptively learns a local confidence threshold by directly estimating the distribution of scores conditional on the input, without relying on any user-specified function classes. This one-shot estimation design further alleviates the limitation of recomputing conformal prediction in Cherian et al. (2024), while empirically providing the localized conformal factuality guarantee. Additionally, the framework is model-agnostic (requiring no access to LLMs' internal parameters or gradients), operates without additional supervision, and offers a practical solution for deploying LLMs in high-risk domains such as financial risk management, medical diagnosis, and legal judgment.

The example shown in Table 1 presents the filtered results of three methods (CF, CC with conditional boosting (CB), and our proposed AggLCF) for LLMs' answers to the question "*Does Vitamin D Help polymorphic light eruptions?*". CF elaborates on the effects of Vitamin D (e.g., its immunomodulatory properties and role in skin health), but the number of sub-claims remains limited. CB expands on CF by explaining the pathogenic mechanism of polymorphic light eruption (i.e.,

noting it is a skin condition triggered by sunlight exposure); however, constrained by the scarcity of sub-claims extractable from a single model's (e.g., GPT-3.5-turbo) original response, CB cannot further increase the count of valid sub-claims. In stark contrast, AggLCF enriches answers by aggregating outputs from multiple LLMs, integrating diverse perspectives such as the symptoms of PMLE (itchy, blistering rashes), Vitamin D's functions, and precautions/recommendations for medication. This diversity, derived from synthesizing viewpoints across multiple models, better caters to users' need for comprehensive information in practical scenarios, making AggLCF more appealing.

## 2 RELATED WORKS, MOTIVATION AND CONTRIBUTIONS

**Conformal Factuality (CF)**: Mohri & Hashimoto (2024) introduced *Conformal Factuality (CF)*, the first framework to provide rigorous, distribution-free factuality guarantees for black-box LLMs. Its core insight lies in redefining LLM factuality as an uncertainty quantification problem: *(I) Sub-claim Decomposition:* As factuality errors often localize to specific claims rather than entire outputs, for an input prompt $P_i$, CF splits the LLM's response $R_i$ into $k_i$ atomic sub-claims $\boldsymbol{C}_i = \{C_{ij}\}_{j=1}^{k_i}$ following with the ground truth label (e.g., binary variables, 1 for true, 0 for false) of each sub-claims $\boldsymbol{W}_i = \{W_{ij}\}_{j=1}^{k_i}$; *(II) Conformity Scores:* CP assigns each sub-claim $C_{ij}$ a confidence score via the scoring function $p(P_i, C_{ij}) \in \mathbb{R}$ which also takes the prompt $P_i$ as input. Define conformity score $S_i := \inf \{\tau \mid \forall C_{ij} \in \mathcal{F}(\boldsymbol{C}_i; \tau), W_{ij} = 1\}$ as the local minimal threshold, where $\mathcal{F}(\boldsymbol{C}_i; \tau) = \{C_{ij} \mid p(P_i, C_{ij}) \geq \tau\} \subseteq \boldsymbol{C}_i$ is the filtered sub-claim set; *(III) Probabilistic Guarantee with Global Threshold:* Based on split conformal prediction (Vovk et al., 2005; Lei et al., 2018), by setting the $\lceil (1-\alpha)(n+1)/n \rceil$-quantile of the conformity scores $\{S_i\}_{i=1}^n$ as the global threshold $\hat{\tau}$ for all input prompts, Mohri & Hashimoto (2024) provides the probabilistic guarantee that the filtered sub-claim $\hat{\mathcal{F}}(\boldsymbol{C}_{n+1}) = \mathcal{F}(\boldsymbol{C}_{n+1}; \hat{\tau})$ contains no false claims with high probabilities of $1 - \alpha$:

$$\mathbb{P}\left(\exists\, C_{(n+1)j} \in \hat{\mathcal{F}}(\boldsymbol{C}_{n+1}) \text{ s.t. } W_{(n+1)j} = 0\right) \leq \alpha, \tag{1}$$

for exchangeable $\{(P_i, R_i, \boldsymbol{C}_i, \boldsymbol{W}_i)\}_{i=1}^{n+1}$. Conformal Factuality provides rigorous factuality guarantees for any black-box LLM, requiring few human annotations. However, it suffers from critical limitations rooted in its global threshold design. Derived from the worst-case set $\{S_i\}_{i=1}^n$, the global threshold $\hat{\tau}$ ensures only marginal guarantee and fails to capture localized validity across heterogeneous inputs (e.g., topics or difficulty levels). Combine with imperfect scoring functions $p(\cdot, \cdot)$, this conservative design induces a "say less to err less" phenomenon, excessively filtering true but low-scoring sub-claims, undermining the completeness and practical utility of generated content.

**Conditional factuality**: While marginal coverage keeps the average error rate of the prediction set below $\alpha$ across all cases, the error rate for certain local regions can still be much higher than $\alpha$, which is unacceptable in high-stakes fields. We therefore seek more localized guarantees, namely, conditional coverage, which is the goal of conditional conformal prediction, i.e.,

$$\mathbb{P}\left(\exists\, C_{(n+1)j} \in \hat{\mathcal{F}}(\boldsymbol{C}_{n+1}) \text{ s.t. } W_{(n+1)j} = 0 \mid P_{n+1}\right) \leq \alpha, \tag{2}$$

Although attractive, conditional coverage (2) is unachievable in finite-sample, distribution-free settings (Lei et al., 2013; Barber et al., 2021). In classical regression and classification, recent studies develop prediction sets that maintain distribution-free marginal guarantees with approximate or asymptotic conditional coverage guarantees (Romano et al., 2019; Chernozhukov et al., 2021; Lei et al., 2018; Lei & Wasserman, 2014; Guan, 2023; Hore & Barber, 2025; Gibbs et al., 2025).

**Conditional conformal for LLMs**: Based upon the conditional conformal prediction framework (Gibbs et al., 2025), Cherian et al. (2024) extended CF to enable conditionally validity. To build an adaptive threshold conditional on the input feature $X_i$ (gained from prompt $P_i$ and response $R_i$) and obtain (approximately) conditionally valid factuality guarantee, Cherian et al. (2024) learns a conditional threshold function over a *user-specified* function class $\mathcal{F}$ (e.g., finite dimensional linear class) that estimates the quantile of the conformity scores. Furthermore, to retain as much of the LLM output as possible while still ensuring validity, Cherian et al. (2024) proposed *Conditional Boosting* that learns a parametric combination of base scorers $p_\theta(\cdot)$ that optimally weights multiple base scoring functions. However, it incurs substantial computation due to repeated scoring, quantile estimation (with randomization), and backpropagation through the differentiable surrogate.

**Motivations:** Mohri & Hashimoto (2024); Cherian et al. (2024) calibrate a *single* LLM, relying on a single LLM to generate limited sub-claims, which misses potential valid sub-claims that the LLM fails to produce. The lack of diversity in responses not only limits sub-claim factuality but also weakens the robustness of extracted features from a single LLM. Although this naturally leads us to consider using multi-LLM to extend sub-claim coverage, directly using existing multi-model aggregation methods faces the problems of semantic redundancy and inconsistency of sub-claims, and these methods fail to provide rigorous factuality guarantees.

These limitations motivate our innovations of:

1. Multi-LLM aggregation: Enlarge sub-claim diversity and provide multi-model aggregation enhanced features on the premise of addressing the semantic redundancy/conflicts of sub-claims.

2. After semantically clustering the sub-claims produced by all large language models, we fuse the resulting cluster-level features to derive a unified confidence score $p$ for each cluster, aiming for this score to align as closely as possible with sub-claim correctness.

3. Lightweight localized conformal factuality: By focusing on score–correctness coherence rather than directly maximizing retained claims in Cherian et al. (2024), we significantly reduce computation while still improving retention indirectly, as a better-calibrated score reduces the number of discarded correct claims at any given error threshold.

**Contributions.** In this paper, our goal is to build a lightweight framework with robust conditional coverage guarantees, while ensuring marginal guarantees with localized factuality control on expanded sub-claims by multi-LLMs. Our main contributions are summarized below:

- **Semantic and logical consistency fusion scoring:** We design a composite confidence function that integrates intra-claim semantic similarity and inter-model consistency. This scoring mechanism enhances discriminative power by capturing both the robustness of individual sub-claims and agreement across multiple models.

- **Localized conformal factuality:** We extend standard conformal prediction in CF to a conditional setting, learning localized thresholds rather than a global one. We prove that this approach controls the factuality error rate under $\alpha$. Numerical results show that our proposed AggLCF improves the number of retained sub-claims.

- **Black-box friendly and lightweight implementation:** Our approach relies solely on model outputs and requires only lightweight post-processing at the sub-claim level. It is compatible with both open- and closed-source LLMs via API calls, ensuring easy portability and reproducibility across tasks and domains.

## 3 Multi-Model Aggregation for Conformal Inference

### 3.1 Problem Formulation

Instead of relying on a single LLM as in Mohri & Hashimoto (2024) and Cherian et al. (2024), for the same question/prompt $P_i$, we simultaneously query $M$ different LLMs and collect responses $\{R_i^{(m)}\}_{m=1}^M$. Each response $R_i^{(m)}$ is decomposed into sub-claims $\boldsymbol{C}_i^{(m)} = \{C_{ij}^{(m)}\}_{j=1}^{k_i^{(m)}}$, yielding a richer dataset:

$$\left\{ D_i = \left( P_i, T_i, \{R_i^{(m)}\}_{m=1}^M, \{\boldsymbol{C}_i^{(m)}\}_{m=1}^M \right) \right\}_{i=1}^n, \tag{3}$$

where $T_i$ is the ground-truth answer to $P_i$. For the $i$-th question, the complete set of sub-claims generated by all models is

$$\mathbf{C}_i = \bigcup_{m=1}^M \boldsymbol{C}_i^{(m)} = \{C_{ij}^{(m)} \mid 1 \le m \le M, 1 \le j \le k_i^{(m)}\}.$$

Our objective is to select a subset of sub-claims that are mutually semantically distinct, while ensuring that the probability of any factual error among these chosen sub-claims remains below a specific

significance level $\alpha$. In other words, given a new question $P_{n+1}$, we construct a filtered set of sub-claims $\hat{\mathcal{F}}(\mathbf{C}_{n+1}) \subset \mathbf{C}_{n+1}$ such that

$$\mathbb{P}\left(\exists\, C_{(n+1)j}^{(m)} \in \hat{\mathcal{F}}(\mathbf{C}_{n+1}) \text{ s.t. } W_{(n+1)j}^{(m)} = 0\right) \leq \alpha \tag{4}$$

where $W_{(n+1)j}^{(m)} = 1$ if $C_{(n+1)j}^{(m)}$ is correct and 0 otherwise, based on the true answer $T_{n+1}$.

## 3.2 PROPOSED METHODOLOGY

Within the full set of sub-claims $\mathbf{C}_i$ made by $M$ models for $i$-th question, it may exist some sub-claims that are semantic redundant or inconsistent. While filtering on the full set can eliminate sub-claims with semantic conflicts, the remaining sub-claims with semantic redundancy will introduce bias into the quantiles. To ensure the filtered sub-claims set $\hat{\mathcal{F}}(\mathbf{C}_i)$ meets our objective in (4), we first group the sub-claims into distinct semantic groups via semantic clustering.

Let $\text{Cluster}(\cdot)$ denote a text clustering algorithm. Applying the clustering algorithm to $\mathbf{C}_i$ yields

$$\mathbf{G}_i = \text{Cluster}(\mathbf{C}_i) = \{\boldsymbol{G}_i^1, \boldsymbol{G}_i^2, \ldots, \boldsymbol{G}_i^{L_i}\}, \tag{5}$$

where $\boldsymbol{G}_i^l = \{G_{ij}^l\}_{j=1}^{d^l}$ denotes the $l$-th cluster of $d^l$ sub-claims obtained from clustering, and $L_i$ is the number of clusters for the $i$-th question. They satisfy

$$\boldsymbol{G}_i^1 \cup \boldsymbol{G}_i^2 \cup \cdots \cup \boldsymbol{G}_i^{L_i} = \mathbf{C}_i, \quad \boldsymbol{G}_i^a \cap \boldsymbol{G}_i^b = \emptyset \quad (\text{for } a \neq b). $$

**Cluster Conformity.** Unlike Mohri & Hashimoto (2024) and Cherian et al. (2024), which filter each sub-claim generated by a single LLM, we calculate a confidence score based on the intra-cluster features of $\boldsymbol{G}_i^l$. The confidence score function $p_i^l$ is a function of $d$ cluster features $f_{i1}^l, f_{i2}^l, \cdots, f_{id}^l$ (e.g., intra-cluster median similarity, nearest-neighbor separation, cluster size). In practice, we train a binary classifier on the training set and treat its predicted probability as the confidence score $p_i^l$, so that the score is as well aligned as possible with the true correctness of each subclaim.

Subsequently, we define the score of cluster set $\mathbf{G}_i$ for $i$-th question as

$$S_i := \inf\left\{\tau \mid \forall\, \boldsymbol{G}_i^l \in \mathcal{F}(\mathbf{G}_i; \tau), W_i^l = 1\right\}, \quad \text{where} \quad \mathcal{F}(\mathbf{G}_i; \tau) = \left\{\boldsymbol{G}_i^l \mid p_i^l \geq \tau\right\} \tag{6}$$

and $W_i^l$ is the correctness indicator assigned by an evaluation function $\text{Eval}(T_i, g_i^l)$, where $g_i^l$ is the representative exemplar selected from the centroids of the cluster $\boldsymbol{G}_i^l$. The details of the evaluation function $\text{Eval}(T_i, g_i^l)$ can be found in Appendix C.1. Since the sub-claims are grouped by semantic similarity, we treat $g_i^l$ as the semantic representative of every sub-claim in its cluster $\boldsymbol{G}_i^l$. Accordingly, the correctness of $g_i^l$ is taken as the correctness of the entire cluster $\boldsymbol{G}_i^l$.

**Localized Conformal Factuality.** Similar to Romano et al. (2019), our proposed localized conformal factuality approach is motivated by the following observation: if the conditional distribution of the cluster scores $S_i$ given the question $P_i$ is know, then the $(1 - \alpha)$-quantile of this distribution can be used as the threshold, which in turn provides an exact conditional coverage guarantee in theory. However, this conditional distribution is unknown in practice. Moreover, since each question $P_i \in P$ is a complex sentence whose encoded representation would yield a high-dimensional vector, directly estimating the conditional distribution of $S_i$ given $P_i$ is infeasible. To address this, we employ feature extraction. Specifically, for each question $P_i$, we extract graph-structured features $X_i$ based on the responses $\{R_i^{(m)}\}_{m=1}^M$ from all large models (feature extraction of $X_i$ are detailed in Appendix C). We then estimate the conditional distribution $f_{S|X}(s \mid x)$ of $S_i$ given $X_i$, and denote $\hat{f}_{S|X}(s \mid x)$ as the estimator.

Directly using the $(1 - \alpha)$-quantile of $\hat{f}_{S|X}(s \mid x)$ as the final threshold $\hat{\tau}$ would not guarantee marginal coverage. Hence, we perform a calibration process. Specifically, we split the data into a training set $D^{\text{tr}}$ and a calibration set $D^{\text{cal}}$. On the training set $D^{\text{tr}}$, we estimate the conditional distribution $\hat{f}_{S|X}(s \mid x)$. On the calibration set, we define

$$E_i = \max\left\{S_i - Q\left(1 - \alpha, \hat{f}_{S|X}(\cdot \mid X_i)\right), 0\right\}, \tag{7}$$

where $Q\left(1 - \alpha, \hat{f}_{S|X}(\cdot \mid x)\right)$ is the $(1 - \alpha)$-quantile of $\hat{f}_{S|X}(s \mid x)$. We then compute

$$\hat{e} = Q\left(1 - \alpha, \frac{1}{n_{\text{cal}} + 1}\left(\sum_{i=1}^{n_{\text{cal}}} \delta_{E_i} + \delta_{\infty}\right)\right), \tag{8}$$

where $\delta_a$ is the point mass measure at $a$, and $n_{\text{cal}}$ is the size of calibration set $D^{\text{cal}}$.

For a new question $P_{n+1}$, the final localized threshold is taken as

$$\hat{\tau}(X_{n+1}) = Q\left(1 - \alpha, \hat{f}_{S|X}(\cdot \mid X_{n+1})\right) + \hat{e}. \tag{9}$$

The filtered set is

$$\hat{\mathcal{F}}(\mathbf{C}_{n+1}) := \left\{\boldsymbol{G}_{n+1}^l \in \mathbf{G}_{n+1} \mid p_{n+1}^l \geq \hat{\tau}(X_{n+1})\right\}. \tag{10}$$

Our proposed localized conformal factuality, through the process of estimating the conditional distribution and calibration, enjoys desirable local performance as demonstrated in our experiments in Section 4. Its marginal factuality error rate is theoretically controlled, as established in the following theorem, with the proof provided in Appendix A.

**Theorem 3.1.** *Assume the calibration set size is $n_{cal}$ and all conformity scores $\{E_i\}_{i=1}^{n_{cal}}$ are exchangeable. Then, the set of sub-claims $\hat{\mathcal{F}}(\boldsymbol{G}_{n+1}) = \mathcal{F}(\boldsymbol{G}_{n+1}, \hat{\tau}(X_{n+1}))$ filtered using the threshold $\hat{\tau}(X_{n+1})$ obtained from (9) enjoys the following marginal coverage guarantee:*

$$\mathbb{P}\left(\exists\, G_{n+1}^l \in \hat{\mathcal{F}}(\boldsymbol{G}_{n+1}) \text{ s.t. } W_{n+1}^l = 0\right) \leq \alpha. \tag{11}$$

---

**Algorithm 1** Aggregation Enhanced Localized Conformal Factuality (AggLCF)

---

**Require:** Dataset: $\{D_i = (P_i, T_i, \{R_i^{(m)}\}_{m=1}^M, \{\boldsymbol{C}_i^{(m)}\}_{m=1}^M)\}_{i=1}^n$, significance level $\alpha \in (0, 1)$, Clustering algorithm $\text{Cluster}(\cdot)$, evaluation function $\text{Eval}(\cdot, \cdot)$, Test question with $M$ LLMs' responses: $(P_{n+1}, \{R_{n+1}^{(m)}\}_{m=1}^M, \{\boldsymbol{C}_{n+1}^{(m)}\}_{m=1}^M)$

**Ensure:** Filtered sub-claim set $\hat{\mathcal{F}}(\mathbf{G}_{n+1})$ for test input $(P_{n+1}, R_{n+1}^{(m)})$

1: **for** each $D_i = (P_i, T_i, \{R_i^{(m)}\}_{m=1}^M, \{\boldsymbol{C}_i^{(m)}\}_{m=1}^M)$ in $\{D_i\}_{i=1}^n$ **do**
2:     Extract features $X_i$ from $(P_i, \{R_i^{(m)}\}_{m=1}^M)$.
3:     Compute the union of multi-model sub-claims: $\mathbf{C}_i = \bigcup_{m=1}^M \boldsymbol{C}_i^{(m)}$.
4:     Apply clustering algorithm $\text{Cluster}(\mathbf{C}_i)$ to get clustered sub-claim set $\mathbf{G}_i$
5:     Compute a confidence score $p_i^l$ for each cluster $\boldsymbol{G}_i^l$ in $\mathbf{G}_i$.
6:     Validate the correctness of centroid sub-claim $g_i^l \in \boldsymbol{G}_i^l$ by $W_i^l = \text{Eval}(T_i, g_i^l)$.
7:     Calculate the overall score $S_i$ of clustered set $\mathbf{G}_i$ for sample $D_i$ using Eq. (6).
8: **end for**
9: Split the training data $\{D_i\}_{i=1}^n$ (with corresponding $S_i$ and $X_i$) into two disjoint subsets: training set $\mathcal{D}_{tr}$ and calibration set $\mathcal{D}_{cal}$.
10: Use $\mathcal{D}_{tr}$ to train the conditional distribution estimator $\hat{f}_{S|X}(s \mid x)$, and compute conformity scores for $\mathcal{D}_{cal}$ using Eq. (7).
11: Calculate the empirical quantile of the conformity scores via Eq. (8).
12: For the test sample $(P_{n+1}, \{R_{n+1}^{(m)}\}_{m=1}^M, \{\boldsymbol{C}_{n+1}^{(m)}\}_{m=1}^M)$:
    1. Repeat the same step in lines 2 to 5 to obtain $X_{n+1}$ and score $p_{n+1}^l$ for each $\boldsymbol{G}_{n+1}^l \in \mathbf{G}_{n+1}$.
    2. Determine the final localized threshold using Eq. (9).
    3. Construct the filtered sub-claim set $\hat{\mathcal{F}}(\mathbf{G}_{n+1})$ using Eq. (10).

---

### 3.3 CONSTRUCTION OF THE CONFIDENCE SCORE $p_i^l$

Our filtering rule (10) is most effective when the confidence score $p_i^l$ is aligned with the correctness of each sub-claim $W_i^l$: if, within a question, all correct sub-claims have larger scores than all incorrect ones, then the threshold in (6) retains only correct sub-claims and filters all incorrect ones out.

After clustering, we extract cluster-level features $\boldsymbol{f}_i^l = (f_{i1}^l, \ldots, f_{id}^l)$ (e.g., intra-cluster median similarity, separation to nearest neighbor) for each cluster $\boldsymbol{G}_i^l$, and train a binary classifier on the *training set* to map $\boldsymbol{f}_i^l$ to a correctness probability,

$$p_i^l = \mathbb{P}(W_i^l = 1 \mid \boldsymbol{f}_i^l). \tag{12}$$

This learning step uses no calibration or test data, and serves only to produce a score that tracks correctness. We then perform a single split–conformal calibration on the *calibration set* to estimate a global quantile threshold. Because calibration and test examples are exchangeable, the assumptions of Theorem 3.1 hold, so the procedure preserves the desired *marginal* factuality guarantee while benefiting from a score $p_i^l$ that is well aligned with $W_i^l$.

In a nutshell, learn $p_i^l$ on train (to increase true-retention at any fixed error level), calibrate once on a disjoint set (to maintain exchangeability and marginal validity), and evaluate on test. Empirically, the localized thresholds induced by our learned scores achieve coverage close to $1 - \alpha$ across question types, while the marginal guarantee is retained by construction. Compared with prior work that relies on model "confidence" or token-level uncertainty, our design is (a) more hallucination-resistant (no self-assessment), (b) more robust to prompt/temperature variation (grounded in pairwise similarity and cluster geometry), and (c) more interpretable (each feature has a clear geometric or reliability meaning).

## 4 EXPERIMENTS

**Datasets and Evaluation Metrics.** Following the same dataset split procedure in Cherian et al. (2024), we conduct experiments on five medical QA datasets in the MedLFQA benchmark (Jeong et al., 2024, Medical Long-Form Fact-based Question-Answering), which is a consolidated and cleaned collection derived from four classic medical QAs. A reference answer and author-annotated key claims accompany each question. The benchmark details are shown in Table 2. We split the data into Training set $D^{tr}$ (50%), Calibration set $D^{cal}$ (20%), and Test set $D^{test}$ (30%) using stratified sampling across all four QA datasets to ensure consistent category distribution in each subset.

Among the five different data sources, erroneous responses to challenging or rare questions, such as those on LiveQA and MedicationQA, pose higher risks, particularly those involving precise medication usage. Thus, we evaluate the marginal coverage and local conformal factuality of each method on sub-datasets by measuring error frequency (i.e., the proportion of test set samples containing incorrect sub-claims in the filtered output). On the premise of ensuring coverage, we primarily assess the quality of the filtered sub-claims of these methods using the number of retained sub-claims. A larger average number of retained sub-claims indicates that, under the guarantee of factuality control, the method can retain more diverse answers to the problem.

Table 2: MedLFQA Statistics

| Data Source | Number of Entries | Description |
|---|---|---|
| HealthSearchQA | 3,047 | Web health search Q&A pairs |
| K-QA (Golden) | 201 | Clinical Q&A pairs with **doctor-annotated** answers |
| K-QA (Silver) | 876 | Clinical Q&A pairs with **LLM-generated** answers |
| LiveQA | 100 | real-time health Q&A pairs |
| MedicationQA | 627 | medication consultation Q&A pairs |

**Comparison baselines.** We denote the baseline method (Mohri & Hashimoto, 2024) that first applies Conformal Inference to Factuality in large language models as **CF**. Following this work, the conditional conformal method proposed by Cherian et al. (2024) introduces two approaches to enhance the sub-claim retention rate: Conditional Boosting (denoted as **CB**) and Adaptive Level. Specifically, we denote the original method without CB as Conditional Conformal inference (**CC**), and the method incorporating CB as **CC+CB**. Following the settings in Cherian et al. (2024), gpt-3.5-turbo is used for response generation to reproduce prior SOTA performance. The objective of this paper is to maximize the sub-claim retention rate under the premise of strict marginal coverage guarantees. Notably, the Adaptive Level method proposed in Cherian et al. (2024) enhances the number of retained claims by relaxing the coverage guarantee of $\alpha$. Therefore, the Adaptive Level

method falls outside the scope of this paper. Results with mean and standard deviation are reported for all methods, obtained from 50 individual trials.

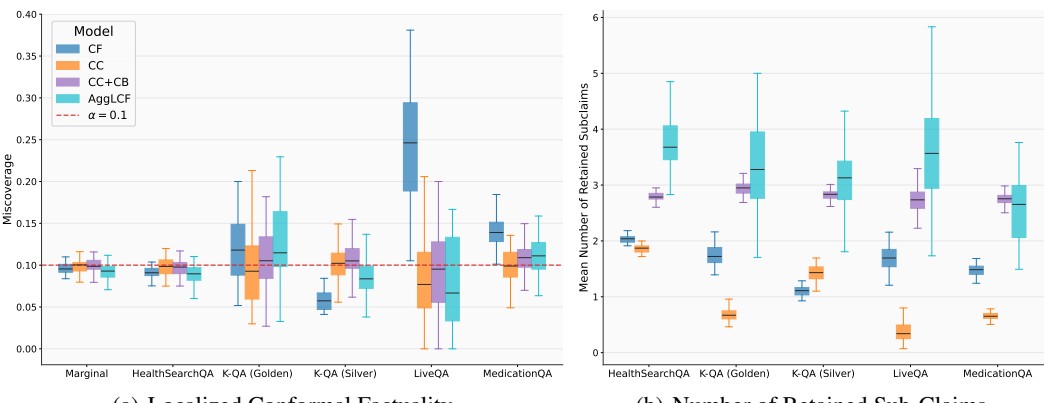

(a) Localized Conformal Factuality

(b) Number of Retained Sub-Claims

Figure 1: Comparison of CF (Mohri & Hashimoto, 2024), CC, CC+CB (Cherian et al., 2024), and AggLCF (ours) on MedLFQA benchmarks with the target level $\alpha = 0.1$.

**Multi-Model Response Generation and Subclaim Decomposition.**  For each question in the test set, we query $M = 10$ often-used LLMs to generate responses in parallel: moonshot-v1-8k, llama2-7b, llama2-13b, llama3-8b-instruct, llama3-70b-instruct, qwen-plus, qwen-32b, deepseek-v3, deepseek-r1, gpt-3.5-turbo-0613.  Model parameters are uniformly set to temperature= 0.2, top-$p = 0.95$.  We use the same prompts with GPT-4o in Cherian et al. (2024) to decompose each response into $k \in \{3, 4, 5, 6\}$ sub-claims, averaging $k = 5.915$.  The decomposition prompts are detailed in Appendix C.

**Performance Comparision**    For each sub-dataset within the MedLFQA benchmark, we compare the **miscoverage** and **average number of retained sub-claims** of each method. As shown in Figure 1(a), all methods control the miscoverage required by the target level $\alpha = 0.1$. On the rarer LiveQA sub-dataset and the more challenging MedicationQA sub-dataset, CF failed to provide local coverage, while both CC/CC+CB and AggLCF demonstrated comparable local factuality control. CB achieves a significant improvement on CC in the number of retained sub-claims (see Figure 1(b)) at the cost of repeatedly recalculating conformal predictions during the optimization. In contrast, through multi-model aggregation, AggLCF achieves a noticeably higher number of retained sub-claims than CB **without heavy computational overhead**. Specifically, AggLCF enriches responses with a significantly larger number of valid sub-claims than other methods on both the majority sub-dataset (HealthSearchQA, $n = 3047$) and the rare sub-dataset (LiveQA, $n = 100$), offering robust enhancement.

We evaluate the error rates across different target levels $\alpha$ and compare the average number of sub-claims retained by each method at each level.  The left panel of Figure 2 demonstrates that AggLCF consistently maintains strict edge coverage.  Since the output of a single model can be divided into at most 6 semantically distinct sub-arguments, methods relying on a single model can retain only up to 6 sub-arguments even when the constraint on $\alpha$ is relaxed. In contrast, AggLCF can continuously enrich the content of the responses.  The right panel of Figure 2 demonstrates that AggLCF consistently increases the number of retained sub-claims, while providing strict guarantees on overall coverage (as shown in the left panel).

**Importance of Cluster-level features.**    As we discussed in Section 3.3, we extracted five features from clusters we obtained after multi-model aggregation. To demonstrate the importance of integrating these geometrical features in improving the number of retained sub-claims rather than relying on a single feature, we compare the number of retained sub-claims when each feature is used individually versus when all clustering features are integrated.

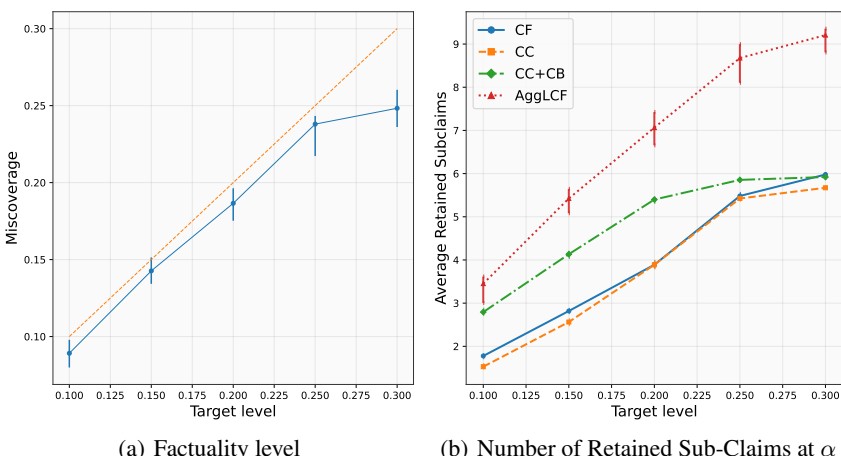

(a) Factuality level         (b) Number of Retained Sub-Claims at $\alpha$

Figure 2: Sensitivity analysis with respect to the target level $\alpha$. The left figure illustrates the performance of AggLCF under various $1 - \alpha$ coverage requirements. The figure on the right compares the average number of retained sub-claims among CF, CB, and AggLCF for each target level.

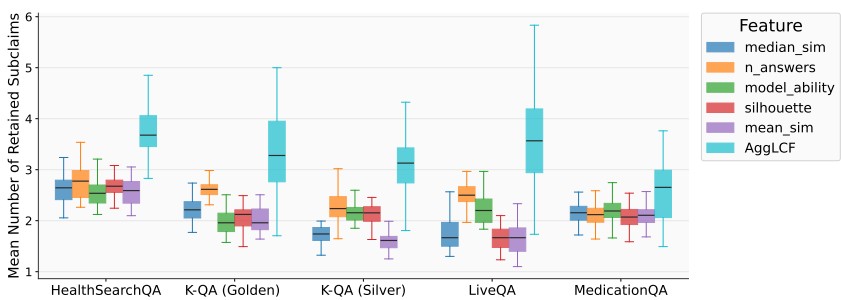

Figure 3: Sensitivity analysis on the construction of the confidence score $p_i^l$. As shown in the figure, combining multiple cluster-level features into a single confidence score substantially increases the number of retained subclaims, compared to using any single feature alone.

As illustrated in Figure 3, when each of the five clustering features is used alone as a score for evaluating correctness, the number of retained sub-claims is relatively similar, indicating these individual cluster-level features are empirically unable to align well with the correctness of the response. In contrast, when cluster-level features are used together, the number of retained sub-claims is significantly improved. This indicates that the integrated cluster-level features can better align with the correctness of answers.

## 5 CONCLUSION

This paper proposes a novel method to provide factuality guarantees for the outputs of LLMs while maximizing the number of retained sub-claims. Specifically, AggLCF enhances the diversity of filtered model responses by performing semantic clustering on the aggregated output of multiple models. Unlike prior approaches, AggLCF directly estimates the conditional distribution on input features, achieving robust approximate local coverage performance without relying on user-specified function classes. By leveraging cluster features that can be extracted in an offline manner, AggLCF constructs a scoring metric with better alignment to the correctness of claims, thereby effectively increasing the number of filtered sub-claims. Extensive experiments demonstrate that AggLCF consistently improves the number of retained sub-claims under various target level requirements while providing rigid marginal coverage guarantees and localized conformal factuality.

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

## A  PROOF OF THEOREM 3.1

*Proof.* Let

$$S_{n+1} := \inf \left\{ \tau \mid \forall\, G_{n+1}^\ell \in \mathcal{F}(G_{n+1}, \tau), W_{n+1}^\ell = 1 \right\}, \tag{13}$$

where

$$\mathcal{F}(G_{n+1}; \tau) = \left\{ G_{n+1}^\ell \mid p_{n+1}^\ell \geq \tau \right\}. \tag{14}$$

Consider the event

$$\left\{ \exists\, G_{n+1}^{\ell_0} \in \hat{\mathcal{F}}(G_{n+1}),\ \text{s.t.}\ W_{n+1}^{\ell_0} = 0 \right\} = \left\{ \exists\, G_{n+1}^{\ell_0}, p_{n+1}^{\ell_0} \geq \hat{q}_{1-\alpha}(X_{n+1}) + \hat{e} \wedge W_{n+1}^{\ell_0} = 0 \right\}. \tag{15}$$

Define the set

$$A = \left\{ \tau \mid \forall\, G_{n+1}^\ell, p_{n+1}^\ell \geq \tau, W_{n+1}^\ell = 1 \right\}. \tag{16}$$

For any $s < \hat{q}_{1-\alpha}(X_{n+1}) + \hat{e}$, we have

$$p_{n+1}^{\ell_0} \geq s \wedge W_{n+1}^{\ell_0} = 0, \Rightarrow s \notin A \tag{17}$$

This implies $\inf A \geq \hat{q}_{1-\alpha}(X_{n+1}) + \hat{e}$, i.e., $S_{n+1} \geq \hat{q}_{1-\alpha}(X_{n+1}) + \hat{e}$. Therefore,

$$E_{n+1} = \max\{S_{n+1} - \hat{q}_{1-\alpha}(X_{n+1}), 0\} \geq \hat{e}. \tag{18}$$

Thus, we have the event inclusion:

$$\left\{ \exists\, G_{n+1}^\ell \in \hat{\mathcal{F}}(G_{n+1}),\ \text{s.t.}\ W_{n+1}^\ell = 0 \right\} \subset \{ E_{n+1} \geq \hat{e} \}. \tag{19}$$

We can then bound the desired probability:

$$\mathbb{P}(\exists\, G_{n+1}^\ell \in \hat{\mathcal{F}}(G_{n+1}),\ \text{s.t.}\ W_{n+1}^\ell = 0) \tag{20}$$
$$\leq\ \mathbb{P}(E_{n+1} \geq \hat{e}) \tag{21}$$
$$=\ \mathbb{E}[\mathbb{I}\{E_{n+1} \geq \hat{e}\}] \tag{22}$$
$$=\ E\left[ \frac{1}{n_{\text{cal}} + 1} \sum_{i=1}^{n_{\text{cal}}} \mathbb{I}\left\{ E_i \geq Q\left( 1-\alpha, \frac{1}{n_{\text{cal}}+1} \sum_{j=1}^{n_{\text{cal}}} \delta_{E_j} + \delta_\infty \right) \right\} \right]$$
$$\leq\ \alpha \tag{23}$$

By the exchangeability of $\{E_i\}_{i=1}^{n_{\text{cal}}+1}$ (where $E_{n+1}$ is exchangeable with the calibration scores), and by the construction of $\hat{e}$ as the $(1-\alpha)$-quantile of the empirical distribution of the calibration scores augmented with $\infty$, standard conformal prediction arguments yield:

$$\mathbb{E}[\mathbb{I}\{E_{n+1} \geq \hat{e}\}] \leq \alpha. \tag{24}$$

This completes the proof. □

## B  IMPLEMENTATION DETAILS OF AGGLCF

### B.1  CLUSTERLLM METHOD

CLUSTERLLM (Zhang et al., 2023) augments standard embedding-based text clustering with lightweight LLM judgments to improve cluster reliability without labels. We first encode texts with a pretrained sentence embedder to generate text embeddings and run a preliminary algorithm to get provisional clusters. We then identify 'ambiguous' points (high-entropy/low-margin cases) and form triplets—(anchor, choice$_1$, choice$_2$), on which an LLM is queried to decide which choice is semantically closer to the anchor. These sparse LLM comparisons act as higher-order semantic constraints that correct assignments and sharpen boundaries, especially for long-tail or subtle topics. Unlike (Zhang et al., 2023), our goal is to obtain more stable and trustworthy semantic clusters rather than fine-tuning the embedder model.

### B.2 CONDITIONAL DISTRIBUTION ESTIMATION METHOD

When estimating the conditional distribution of $S \mid X$, we use Engression (energy regression) (Shen & Meinshausen, 2025) to estimate $P(S \mid X)$. Given uniform noise $\varepsilon \sim \text{Unif}(0, 1)$, construct a generative mapping

$$S = g_\theta(X, \varepsilon) \tag{25}$$

and minimize the negative energy score

$$L(\theta) = \frac{1}{n} \sum_{i=1}^{n} \left[ \frac{1}{m} \sum_{j=1}^{m} \|Y_i - g_\theta(X_i, \varepsilon_{i,j})\| - \frac{1}{2m(m-1)} \sum_{j=1}^{m} \sum_{j'=1}^{m} \|g_\theta(X_i, \varepsilon_{i,j}) - g_\theta(X_i, \varepsilon_{i,j'})\| \right], \tag{26}$$

where $\varepsilon_{i,j}, \varepsilon_{i,j'} \sim \text{Unif}(0, 1)$. The first term enforces alignment between generated samples and the observed data, whereas the second term regularizes dispersion by promoting pairwise differences among samples. Minimizing this objective jointly improves the fidelity and the diversity of the estimated conditional distribution. After obtaining $\hat{\theta}$, by repeatedly sampling $\varepsilon_j$ and computing $g_{\hat{\theta}}(x, \varepsilon_j)$ on fixed $x$, we can obtain Monte Carlo samples of $P(S \mid X = x)$, and thereby estimate

$$\hat{q}_\alpha(x) = \text{Quantile}_\alpha \{g_{\hat{\theta}}(x, \varepsilon_j)\}_{j=1}^{M}. \tag{27}$$

Due to the use of additive noise and direct matching of the full distribution, Engression theoretically possesses better extrapolation capabilities than traditional regression.

For comparison, we also experimented with two classical quantile regression methods—linear quantile regression and gradient boosting quantile regression to estimate the conditional distribution. As illustrated in the figure 4, the engression-based approach produces a markedly stronger localized-coverage effect than either of the standard quantile-regression baselines.

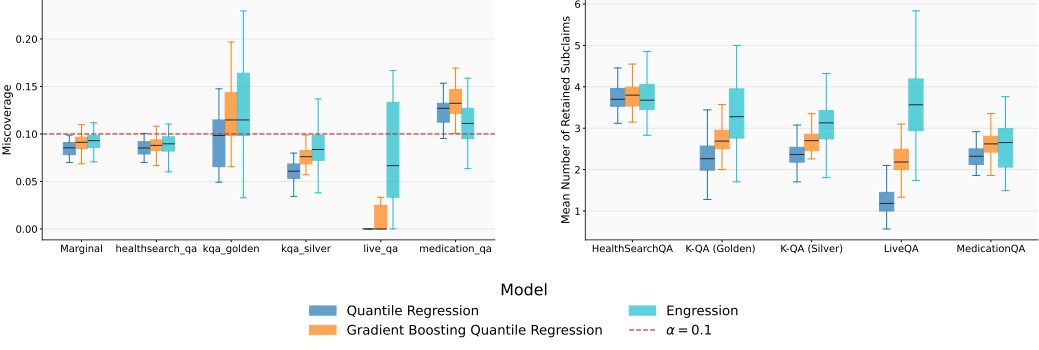

Figure 4: Evaluation of different conditional distribution estimator. Left: marginal and per-question-type miscoverage compared to the nominal error rate (dashed red line at $\alpha = 0.1$), showing how each quantile estimator: linear QR , gradient boosting QR, and engression based QR. Right: corresponding average number of sub-claims retained per question, illustrating the trade-off between retention and error control. Engression yields the most stable, localized miscoverage near the nominal level while preserving the greatest subclaim retention.

## C EXPERIMENT DETAILS

**Feature extraction of $X_i$.** Following Lin et al. (2024), we build three similarity graphs per question (Jaccard, NLI–entailment, NLI–contradiction) and extract, on each graph, the first three non-zero Laplacian eigenvalues (EigV), the average weighted degree (Deg), and the average node eccentricity (Ecc), yielding 15 initial features. We then remove near–zero-variance (degenerate) features and append group indicators, resulting in a 17-dimensional feature vector for downstream modeling.

**Vectorization and Initial Clustering.**

1. **Encoding.** We use `S-BioBERT-snli-nli-stsb` (Deka et al., 2021) to obtain 768-d sentence embeddings. Vectors are mean-pooled over tokens and $\ell_2$-normalized so that cosine similarity is meaningful. The encoder is kept fixed (no fine-tuning) for stability and reproducibility.

2. **Clustering.** We apply HDBSCAN (McInnes et al., 2017) for initial grouping; across questions, the number of discovered clusters averages **10.07** per question (mean over questions).

**Cluster-level features extraction**  After clustering, we derive cluster-level, model-agnostic features directly from the grouped generations, rather than from LLM self-reported confidence. Specifically: (i) the intra-cluster mean similarity and (ii) the intra-cluster median similarity, which quantify cohesion while reducing sensitivity to outliers; (iii) cluster size (the number of models supporting the sub-claim), capturing consensus strength; (iv) the silhouette coefficient, measuring separability from neighboring clusters; and (v) a within-cluster model-ability score, obtained by estimating each base model's overall competence and averaging these scores within the cluster. All five signals are computed offline from the clustering outputs and metadata, without querying the LLM for self-confidence.

**Construction of the confidence score $p_i^l$.**  After extracting cluster-level features $\boldsymbol{f}_i^l = (f_{i1}^l, \ldots, f_{i5}^l)$, we learn a data-driven confidence score $p_i^l$ aligned with sub-claim correctness. Concretely, using the *training split only*, we fit a CATBOOSTCLASSIFIER (Prokhorenkova et al., 2018) that maps each cluster's feature vector to a probability of correctness,

$$p_i^l = \mathbb{P}\big(W_i^l = 1 \mid \boldsymbol{f}_i^l\big).$$

We optimize binary logloss with early stopping (monitoring AUC) and address class imbalance via `scale_pos_weight` (set to $N_{\text{neg}}/N_{\text{pos}}$ for each split). CatBoost models the non-linear patterns in these features and outputs reliable probabilities, which we take as the confidence score $p_i^l$. Importantly, no calibration or test data are used in this step, preserving exchangeability for subsequent conformal calibration and ensuring valid marginal coverage.

## C.1 PROMPTS

**Subclaim decomposition.**  Using GPT-4o, each response is decomposed into $k \in \{3, 4, 5, 6\}$ sub-claims, averaging $k = 5.915$. The decomposition instruction is as follows:

```
Please divide the following medical question response into
individual sub-claims with the following requirements:
1. Strictly divide the response into 3-6 sub-claims.
2. Each subclaim must strictly be based on the provided response
content, without adding any new information.
3. Remove any non-informative content (e.g., transitional sentences,
repetitive statements) and retain only the key points.
4. Each subclaim must be concise, focusing on one specific medical
fact, treatment, condition, or recommendation.
5. Ensure there is no unnecessary overlap between sub-claims.
```

**CLUSTERLLM correction.**  We first compute the clustering entropy for each sample. Then, we select the top 20% high-entropy triplets. Finally, we call GPT-4o to judge similarity and update labels and obtain stable clusters. The reclassification instruction is as follows:

```
Given a question and two candidate answers, reply with "Option 1"
or "Option 2" for the answer more semantically related to the question.

Question: {anchor}
Option 1: {choice1}
Option 2: {choice2}
```

**Sub-claim correctness** $W_i^l$. To assess how well each generated subclaim is supported, we prompt the LLM as follows:

```
You will be given a list of sub-claims and a text passage.
For each claim, determine whether the passage Supported, Refuteed,
or provides NotEnoughInfo regarding the claim.
Return your judgments as a JSON array, maintaining the same order
and length as the input claims.
The claims are: [...] The text is: [...]
```

**Experiment Setting of Conditional Conformal Method** Following the framework of Cherian et al. (2024), we construct four claim-level scores used in CC: a frequency score, obtained by averaging the cosine similarity between the target claim and answers from $M$ auxiliary models; self-evaluation, token-level log probability, and an ordinal score, all computed exactly as in Cherian et al. (2024). For CC, we take the frequency score itself as the confidence score, whereas for CB, we linearly combine the frequency score with the other three scores to produce a boosted output. For feature extraction we restrict the function class to be linear and use covariates of the number of characters in the prompt, the number of characters in the response, the mean frequency score across claims in the response, the standard deviation of those frequency scores, and one-hot group-indicators for the source dataset.

