# OpenReview forum: "AggLCF: Aggregation Enhanced Localized Conformal Factuality for Large Language Models"
_ICLR.cc/2026/Conference — ICLR 2026 Conference Withdrawn Submission_

### Official Review · Reviewer_aVD4 · 2025-10-24

**Soundness:** 2
**Presentation:** 1
**Contribution:** 1
**Rating:** 0
**Confidence:** 2

**Summary:**

* Proposes Aggregation-enhanced Localized Conformal Factuality for LLM long-form QA
* Queries multiple LLMs in parallel and clusters all sub-claims by semantic similiary
* Trains classifer on cluster level features to predict localized confidence threshold for conformal prediction that is adjusted per query to account for input difficulty
* Experiments are conducted on several QA datasets to show better performance than several standard baselines to control sub claim factuality.

**Strengths:**

* The idea of multi-modal aggregation for conformal prediction is well-motivated and interesting; combines multiple LLMs to approximate truth oracle
* The experiments seem to show improvement over baselines across datasets and target levels
* Cluster features be offer interpretable representation for further downstream analysis

**Weaknesses:**

* The overall system is complex and not well justified. It is a "kitchen sink" approach with an exceptionally large number of components and hyperparameters. The pipeline includes: 10 different LLMs , GPT-4 for sub-claim decomposition , S-BioBERT for embedding , HDBSCAN for initial clustering , GPT-4 again for ClusterLLM correction , 17-D graph-Laplacian feature extraction , a CatBoost classifier for confidence scoring , and Engression for conditional density estimation. This high complexity makes the method difficult to reproduce, analyze, and practically deploy, and it is unclear which of these many components are essential.
* High inference cost as need forward passes from multiple LLM, HDBSCAN for clusting, ClusterLLM corrections, GPT-4 API calls, extracting graph Laplacian features, running boosting/engression models.
* The baseline comparison is unfair as they are only run on a single model (GPT-3.5 turbo). It is no suprise that a method with 10x more initial information can retain more valid sub-claims
* The localized factuality claim is weak. The variance of miscoverage is high while the baselines have much lower variance. This does not support claim of "robust local control"
* Labeling pipeline relies on GPT-4, which introduces bias into ground truth to determine if sub-claim is supported or refuted, which is problematic for guaranteeing validity of conformal guarantees.
* Only evaluated on medical QA, unclear whether this framework would transfer to other domains, tasks, and settings. e.g. embedding model is S-BioBERT, pre-trained on biomedical data may not transfer to other datasets
* Lack of strong baselines
* Unclear causal link connection between aggregate feature choices and improvements in calibration performance.

**Questions:**

* What is wall-clock time compared to baselines?
* Could authors provide more ablation to justify each component of the proposed system? Which is the critical parts? What are the critical settings and hyperparameters?
* How many LLMs are necessary? Which LLM models should be chosen? What happens if they are very similar or very different from each other?
* I'm not convinced the exchaneability holds throughout the multiple steps (semantic clustering, re-labeling, feature graphs). How do you ensure the coverage guarantee remains intact?
* Any stress tests where calibration/test come from different sub-datasets?
* What happens when LLMs disagree?
* What about human evals to judge which retained sets are more useful? Why not evaluate with humans to confirm actual utility for more useful or informative responses?
* Will you release code, prompts, and cluster annotations to ensure reproducibility?
* How to measure cluster purity  of correct and incorrect sub claims to guarantee factuality?
* Did you measure label noise from LLM-annotation and how that would degrade performance is acc is lower? Can simulate experiments here.
* What is the rejection rate of AggLCF? How often sub claim is filtered out from all models?
* How is "localized" defined formally? Is it query-conditional?

---

### Official Review · Reviewer_rQaz · 2025-10-26

**Soundness:** 2
**Presentation:** 3
**Contribution:** 2
**Rating:** 4
**Confidence:** 4

**Summary:**

This paper proposes AggLCF (Aggregation enhanced Localized Conformal Factuality), a framework for providing rigorous factuality guarantees for LLM outputs while maximizing information retention. The key innovation is aggregating responses from multiple LLMs and using localized conformal prediction to filter sub-claims with distribution-free marginal coverage guarantees.

**Strengths:**

1. Unlike prior work (CF, CC) that relies on single LLM outputs, AggLCF aggregates diverse responses from M=10 LLMs, potentially increasing the pool of valid sub-claims available for filtering.
2. Theorem 3.1 proves marginal factuality control (P[error] ≤ α), maintaining the distribution-free coverage property of conformal prediction.
3. More efficient than iterative methods (CC+CB), avoiding repeated conformal prediction recomputation.

**Weaknesses:**

1. The paper claims (page 7) that their approach is "(a) more hallucination-resistant (no self-assessment), (b) more robust to prompt/temperature variation (grounded in pairwise similarity and cluster geometry), and (c) more interpretable." None of these claims are validated experimentally. There are no ablations testing: Hallucination rates compared to self-assessment methods; robustness across different prompts or temperature settings, and user studies or interpretability evaluations
2. High variance and inconsistent performance in Figure 1: AggLCF exhibits much larger standard deviation in retained sub-claims compared to all baselines, particularly evident in Figure 1(b). This suggests the method is unstable across trials. In addition, AggLCF appears to meaningfully outperform baselines only on HealthSearchQA (the largest dataset, n=3047). On smaller datasets (K-QA Golden, LiveQA), the error bars heavily overlap. In Figure 1(a), AggLCF actually shows higher miscoverage than CC on K-QA (Golden) and MedicationQA, contradicting claims of superior local control.
3. The paper uses exactly 5 cluster-level features for the confidence score, but provides **no ablation on feature count**. How do we know 5 is sufficient or necessary? The 17-dimensional input feature vector Xi** (Appendix C) also appears arbitrary (3 graph types × 5 statistics + 2 indicators). No sensitivity analysis is provided.
4. Figure 3 shows individual features perform poorly, indicating the score relies on complex, opaque feature interactions learned by CatBoost.
4. While claiming to be "lightweight," querying M=10 LLMs in parallel has non-trivial API costs ($$$) compared to single-model baselines and potential latency issues in production settings
5. The scope of evaluation is limited. Only medical QA domain (MedLFQA) was tested, and there is no analysis of which LLMs contribute most to diversity (are all 10 necessary?). No ablation on number of models M (what happens with M=3, 5, 20?)
6. Limited theoretical analysis of local coverage. Theorem 3.1 only provides marginal guarantees. While Figure 1(a) shows empirical local coverage, there's no theoretical characterization of when/why Engression-based conditional distribution estimation provides approximate local validity.

**Questions:**

1. Why is the variance so high for AggLCF? This is a major practical concern. What causes the instability across runs?

2. Is the improvement statistically significant? Can you provide confidence intervals or hypothesis tests showing AggLCF significantly outperforms baselines on K-QA, LiveQA, and MedicationQA (not just HealthSearchQA)?

3. Why does AggLCF have higher miscoverage than CC on some datasets? This contradicts the claimed advantages.

4. How many features are sufficient? Ablation needed on: (a) number of cluster features, (b) which cluster features are most important, (c) sensitivity to Xi feature extraction choices.

5. What is performance with M=2, 3, 5 vs M=10? Is there a point of diminishing returns?

6. Can you provide experiments showing hallucination-resistance, prompt/temperature robustness, and interpretability?

7. What are the actual wall-clock time and API costs vs single-model baselines?

8. Why does Engression outperform quantile regression? (Figure 4) Is this specific to your feature distribution or generally true?

9. Can you test on other domains (e.g., legal, financial, general knowledge)?

---

### Official Review · Reviewer_Fhoz · 2025-10-30

**Soundness:** 3
**Presentation:** 2
**Contribution:** 3
**Rating:** 6
**Confidence:** 4

**Summary:**

The paper proposes AggLCF, a framework that enhances factuality calibration for long-form QA by combining multi-LLM aggregation with localized conformal thresholds. It collects answers from several LLMs, clusters semantically similar sub-claims, computes cluster-level confidence features, and learns a calibrated confidence estimator with CatBoost. A conditional quantile model is used to obtain localized thresholds with formal marginal coverage guarantees. Experiments on MedLFQA show improved factuality coverage and higher valid-claim retention compared to existing Conformal Factuality and Conditional Conformal methods.

**Strengths:**

1. The idea of combining multi-LLM aggregation with localized conformal calibration is novel and practically meaningful for factual QA.

2. The framework is well-structured, with clear modular design, theoretical guarantees, and detailed ablations (feature importance, α-sensitivity, etc.).

3. The paper is clearly written, with transparent dataset splits, implementation details, and public reproducibility information.

**Weaknesses:**

1.  AggLCF aggregates 10 models, while baselines use only one LLM. Some improvements may stem from richer candidate pools rather than calibration alone. A cost-matched comparison is needed.

2.  Factuality labels rely on GPT-4o judgments without human validation. It would strengthen claims to include inter-rater or robustness analysis.

3. Experiments are only on MedLFQA; generalization to open-domain or non-medical QA remains unclear.

4. The cost and latency of multi-LLM aggregation are not reported, which limits practical assessment.

**Questions:**

1. How does AggLCF perform under equal computational budgets or with a single-LLM setting?

2. Can you show conditional or per-topic coverage curves to support the claim of “localized” reliability?

3. How robust are results to different clustering algorithms or factuality evaluators (e.g., retrieval-based or human judges)?

---

### Official Review · Reviewer_SghF · 2025-11-01

**Soundness:** 2
**Presentation:** 3
**Contribution:** 2
**Rating:** 4
**Confidence:** 5

**Summary:**

This paper introduces AggLCF (Aggregation-enhanced Localized Conformal Factuality) - a framework aimed at improving the reliability of large language models (LLMs) in question answering (QA) by leveraging multi-model aggregation and localized conformal prediction. Building on conformal prediction, which provides theoretical finite-sample coverage guarantees, the paper reviews two recent approaches: (1) Conformal Factuality (CF) provides marginal coverage guarantees using a global threshold but is overly conservative and ignores local variations, and (2) Conditional Conformal (CC) learns localized thresholds using a user-defined function class to retain more valid sub-claims but suffers from instability, high computational cost, and sensitivity to the chosen function class.

**Strengths:**

Introduces a multi-model aggregation mechanism to enhance response diversity.  Aggregates multiple LLM responses to the same question to enhance sub-claim diversity and robustness. Clusters semantically similar sub-claims and extracts structured features for localized threshold estimation.

Evaluated on the MedLFQA benchmark, which emphasizes factual correctness in medical QA, an appropriate and challenging testbed. Experiments on the MedLFQA benchmark demonstrate that AggLCF achieves the highest number of valid retained sub-claims and superior factual coverage compared to CF and CC.

**Weaknesses:**

Experiments are restricted to the MedLFQA benchmark, a specialized dataset. Broader evaluation on general-domain QA or multi-domain datasets is needed to establish the claim.

There is a lot of research on conformal methods for UQ and OOD (for example,
Vishwakarma, Harit, Alan Mishler, Thomas Cook, Niccolo Dalmasso, Natraj Raman, and Sumitra Ganesh. "Prune'n Predict: Optimizing LLM Decision-making with Conformal Prediction." In Forty-second International Conference on Machine Learning; Su, Jiayuan, Jing Luo, Hongwei Wang, and Lu Cheng.
"Api is enough: Conformal prediction for large language models without logit-access." arXiv preprint arXiv:2403.01216 (2024);
"Polysemantic Dropout: Conformal OOD Detection for Specialized LLMs" Gupta et. al. EMNLP 2025 https://arxiv.org/abs/2509.04655
Campos, Margarida, António Farinhas, Chrysoula Zerva, Mário AT Figueiredo, and André FT Martins. "Conformal prediction for natural language processing: A survey." Transactions of the Association for Computational Linguistics 12 (2024): 1497-1516.
The discussion of novelty with respect to the existing SOTA needs to be better discussed and empirically validated.

**Questions:**

How does AggLCF ensure localized conformal coverage theoretically, given that it operates without explicit conditional quantile modeling or function class learning?

How sensitive is AggLCF to the choice and number of LLMs used in aggregation?

Could you clarify how structured features are derived from aggregated sub-claims?

Have you evaluated AggLCF in domains beyond medical QA, such as finance or law, to assess its adaptability to varying claim structures and factual baselines?

How does AggLCF compare with retrieval-augmented generation (RAG) or self-consistency-based approaches in factual coverage and efficiency?

---

### Note · Authors · 2025-11-13

I have read and agree with the venue's withdrawal policy on behalf of myself and my co-authors.